# "The added value of 18f-FDG PET/CT in the assessment of onset and steroid resistant polimyalgia rheumatica"

Patricia Moya-Alvarado [1]*, Alejandro Fernandez Leon[2], Maria Emilia Corica[3], Valle Camacho Marti[2], Diego Alfonso López-Mora[2], Ivan Castellví[1], Hèctor Corominas[1]

1 Department of Rheumatology, Hospital Santa Creu i Sant Pau, Barcelona, Spain, 2 Department of Nuclear Medicine, Hospital de la Santa Creu i Sant Pau, Barcelona, Spain, 3 Department of Rheumatology, Hospital José N. Lencinas, Mendoza, Argentina

* pmoyaa@santpau.cat

**Data Availability Statement:** All relevant data are within the manuscript and its Supporting Information files.

## Abstract

PMR is a common inflammatory rheumatic disease. Although its clinical characteristics are fully recognized, no specific test for its diagnosis has been established to date. Several studies have described a wide variety of diseases that present with polymyalgic symptoms. A $^{18}$FDG-PET/CT scan could help to deal with these differential diagnoses. The goal of our study is to describe the findings of the $^{18}$FDG-PET/CT scan in a cohort of PMR patients and to detail how the $^{18}$FDG-PET/CT scan improves accuracy when diagnosing other underlying conditions. This cross-sectional study enrolled patients with a diagnosis of PMR who underwent to a $^{18}$FDG-PET/CT scan to rule out other diagnosis. The $^{18}$FDG-PET/CT scan was performed either following clinical criteria at the onset of clinical symptoms or when the patient became PMR steroid resistant. Patients' demographic, clinical and analytical data at the moment of the $^{18}$FDG-PET/CT scan were recorded. The final diagnosis was confirmed according to clinical judgement. A total of 103 patients with PMR were included. In 49.51% of patients, the $^{18}$FDG-PET/CT scan was ordered to study resistance to steroid therapy. The final diagnoses of patients were PMR in 70.9% patients, large vessel vasculitis in 15.5%, neoplasms 4.8% and another diagnosis in the rest. The $^{18}$FDG-PET/CT scan is a very useful technique for the study of Polymyalgia Rheumatica, not only to help in the diagnostic process, but also due to its role in the identification of a variety of PMR-like patrons.

## Introduction

Polymyalgia rheumatica (PMR) is the most common inflammatory rheumatic disease in patients over 50 years of age. Although the cause of PMR remains unknown, evidence consistently suggests a multifactorial etiology leading to an immunomediated process [1,2]. Although PMR's clinical characteristics are fully recognized, no specific test for its diagnosis has been confirmed to date. Several studies have described a wide variety of diseases that present with polymyalgic symptoms [2–6] and it is important to rule out all these processes, since their therapy and prognosis differ widely from the classic therapy used for PMR. Moreover, an

**Funding:** The authors received no specific funding for this work.

**Competing interests:** The authors have declared that no competing interests exist.

association between PMR and giant cell arteritis (GCA) has been described. Approximately 50% of patients with GCA present with the clinical symptoms of PMR [7]. The percentage of patients with PMR who present GCA ranges from five to thirty percent depending on the series [3,8]. In one population-based study, a temporal artery biopsy yielded positive histologic findings of GCA in 9% of patients presenting with typical PMR without any clinical manifestation of GCA [9].

Imaging tests could help in these differential diagnoses. These tests have gained significant prominence since the American College of Rheumatology (ACR) and the European League Against Rheumatism (EULAR) launched new classification criteria that take ultrasonography findings into account [10]. Other imaging techniques that could help in the differential diagnosis of PMR are magnetic resonance imaging (MRI) and 18-fluorodeoxyglucose positron emission tomography ([18]FDG-PET/CT). [18]FDG-PET/CT is a diagnostic imaging technique that measures metabolic activity by locating and quantifying glucose consumption. It has been used to diagnose and monitor neoplastic processes, though other specialties are now using it to study clinical symptoms such as fever of unknown origin and other processes of an inflammatory nature. The recognized use of a [18]FDG-PET/CT scan to study pathologies characterized by high glucidic metabolism suggests that it may be a promising technique in the differential diagnostic study of patients with PMR. [18]FDG-PET/CT scan findings in PMR may include increased [18]FDG uptake in shoulder, hip and spinous processes [1,10]. A small number of series of published cases also describe subclinical vasculitis and other pathological entities of malignant origin in the [18]FDG-PET/CT scans of patients with PMR. Nevertheless, the conclusive diagnostic relevance of this technique is as yet unknown when performed to complete the study of patients with clinical PMR.

Considering all these factors, the goal of our study was to examine the findings of [18]FDG-PET/CT scans in a cohort of PMR patients and to determine whether this imaging technique offers the clinician the possibility of diagnosing underlying conditions.

## Material and methods

### Patient groups and clinical assessment

A cross-sectional retrospective study was performed in a cohort of PMR patients at the Department of Rheumatology at a tertiary university hospital. Between April 2011 and April 2018, we enrolled patients who had undergone an [18]FDG-PET/CT scan to rule out other diagnoses. PMR was diagnosed by a rheumatologist in accordance with the ACR/EULAR 2012 classification criteria [11] in all patients included in the study. The study population was classified into two groups: 1) patients with onset PMR (PMR_os), and 2) patients with PMR who were refractory to glucocorticoid therapy (PMR_sr). The first group was defined as patients who were visited in the rheumatology department during the initial diagnostic process or who underwent a [18]FDG-PET/CT scan within six weeks after the diagnosis of PMR. The second group was defined as patients with PMR who did not respond to conventional treatment with steroids or patients who flared when the dose was below 7.5 mg. The criteria for performing the [18]FDG-PET/CT scan were: 1) PMR_os: all patients who presented an onset PMR and were attended by one of the three main investigators (JM, EC, PM); and, 2) PMR_sr: all patients with PMR who were refractory to glucocorticoid therapy independently of which physician attended them. We excluded patients who had a history of neoplasia before the [18]FDG-PET/CT scan was performed.

Upon admission to the study, we collected patients' demographic, clinical and laboratory data. [18]FDG-PET/CT scan variables collected were: the joint involved (shoulders, hips, peripheral joints) and aorta, PET/CT standardized uptake values (SUV), and the tissue to

background ratio (TBR) of each location according to the average activity of the vena cava. [18]FDG-PET/CT images were analyzed by a qualified nuclear medicine specialist.

The final confirmed diagnosis was accepted according to the physician's opinion considering the following combination of data: a) clinical parameters, b) the results of the supplemental examinations (blood test and [18]FDG-PET/CT scan), and c) the outcome of the disease.

The study was approved by the local institutional ethics committee—Comité de Ética de investigación con medicamentos de la Fundació de Gestió Sanitaria del Hospital de la Santa Creu i Sant Pau de Barcelona—(IIBSP-VAS-2013-122) and all patients signed an informed consent form before enrollment.

## [18]FDG-PET/CT assessment

All patients had fasted for at least six hours before [18]FDG administration. After the intravenous injection of [18]FDG (3.7 MBq/kg), they rested for 60 minutes. Images were acquired using a GEMINI PET/CT scanner (PHILIPS Health Systems Amsterdam Holland), integrated with a 64-slice multidetector CT.

PET images were obtained for 180 seconds per position. To attenuate correction and to identify anatomical location, we performed a low-dose, non-contrast-enhanced CT scan (tube voltage: 120kV; effective tube current: 30–100 mA), which included the whole body from the top of the skull to the feet.

## Image analysis

It was difficult to discriminate between synovitis and perisynovitis involvement at [18]FDG uptake sites in the shoulder region using [18]FDG/CT. Therefore, we did not classify shoulder lesions as synovitis or perisynovitis, and all sites thought to correspond to such lesions were regarded as the "shoulder".

We performed [18]FDG PET/CT visual analysis by two experienced Nuclear medicine physicians in order to determine positivity in vasculitis. [18]FDG uptake was assessed in large arteries, proximal joints (shoulders, hips and sternoclavicular joints) and in extraarticular synovial structures (interspinous, ischiogluteal and praepubic bursae). We used the semi-quantitative values of SUV in each vascular and articular structure. Vascular SUVmax measurements were taken drawing a VOI (volume region of Interest) at the level of the most visually active segment of the aorta and at the same level as the cava venous pool. Articular and synovial VOIs were also evaluated in order to measure the SUV value, taking into account the maximum activity uptake area. Vascular activity was normalized using TBR values (target–to–blood pool ratio) to divide the vascular wall SUV by the venous blood pool SUV to correct for blood uptake [12].

## Statistical analysis

All data are expressed as mean ± standard deviation (SD). Categorical variables are presented as absolute frequencies and percentages. Comparisons between independent means were analyzed using the Student's t-test or the Mann-Whitney test using IBM-SPSS, version 25. For the categorical variables, the chi-square test or Fisher's exact test were used as appropriate. Correlations between quantitative variables were analyzed using Pearson's correlation coefficient. The non-parametric test (Kruskal–Wallis) was used for quantitative variables without a normal distribution. The Mann_Whitney U test was used in the post-hoc study. The level of statistical significance was established at 5% (alpha value = 0.05).

## Results

A total of 103 patients with PMR (30 men and 73 women) were included in the study. The average age of the patients was 72.48 ± 9.01 years. In 52 (50.48%) patients, the [18]FDG-PET/CT scan was performed at the onset of the disease based on clinical opinion. In 51 (49.51%) patients, the [18]FDG-PET/CT scan was requested to study PMR refractory to glucocorticoids. The average dose of steroids (prednisone or equivalent) at the time of scanning was 11.78 ± 1.36 mg orally daily. Table 1 shows patients' clinical characteristics at the time of the [18]FDG-PET/CT scan.

The final diagnosis, taking into account the results of the [18]FDG-PET/CT scan, analytical parameters and clinical outcome, was PMR in 73 (70.9%) patients, large vessel vasculitis (LVV) in 16 (15.5%), neoplasms in five (4.8%) and other diagnosis (PMR with elderly-onset rheumatoid arthritis [EORA] (Fig 1), Sjögren's syndrome, small vessel vasculitis and degenerative process) in the rest of the sample.

Table 2 shows the TBR of the various locations according to the final diagnosis. We found significant differences between the TBR of the aorta in LVV patients and the other patients (p< 0.001). In the posthoc study, statistical significance was maintained between the final diagnosis of LVV and the TBR in the aorta (p = 0.002).

When analyzing clinical and analytical variables, we observed significant differences between the group with PMR as a final diagnosis and the patients with other diagnoses for the variables of weight loss (28.8% vs 56.7%; p = 0.013) and amaurosis (1.4% versus 13.3%; p = 0.024).

## [18]FDG-PET/CT in PMR & large vessel vasculitis

Forty (54.8%) of the patients diagnosed with PMR showed increased glucidic metabolism in the shoulder, 27 (37%) in the hips, 23 (31.5%) in other joints, and 34 (46.6%) in one or more

**Table 1. Clinical characteristics of patients at the time of [18]FDG-PET/CT.**

| Patient's clinical features | |
|---|---|
| Women, n (%) | 73 (70.9%) |
| Age (years), mean ± SD | 72.4 ± 9.0 |
| Disease duration (months), mean ± SD | 24 ± 41.9 |
| Asthenia, n (%) | 66 (64.1%) |
| Weight loss, n (%) | 38 (36.9%) |
| Fever, n (%) | 16 (15.5%) |
| Morning stiffness, n (%) | 49 (47.6%) |
| Cervicalgia, n (%) | 71 (68.9%) |
| Shoulder pain, n (%) | 100 (97.1%) |
| Hip pain, n (%) | 89 (86.4%) |
| Arthralgia, n (%) | 26 (25.2%) |
| Headache, n (%) | 23 (22.3%) |
| Amaurosis, n (%) | 5 (4.9%) |
| Jaw claudication, n (%) | 9 (8.7%) |
| Temporal artery tenderness, n (%) | 2 (1.9%) |
| Reactive C protein (mg/L), mean ± SD | 38.0 ± 69.5 |
| Erythrocyte sedimentation rate (mm/s), mean ± SD | 55.9 ± 31.0 |
| Hemoglobin (g/L), mean ± SD | 118.7 ± 24.9 |
| Prednisone dose (MPD), mean ± SD | 11.7 ± 1.3 |
| Duration of PDN before PET/CT (weeks), Median<br> • PMR_os (weeks), Median<br> • PMR_sr (weeks), Median | 13<br>2.7<br>61.1 |

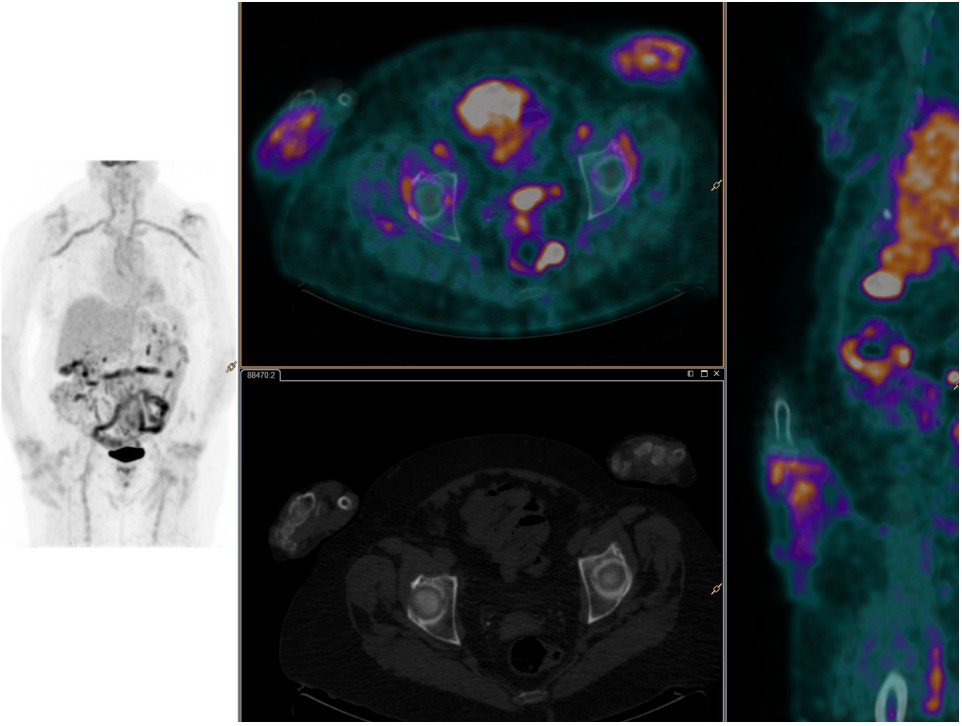

**Fig 1. $^{18}$FDG-PET/CT images.** A 78-year-old woman with diagnostic of elderly onset rheumatoid arthritis [EORA]. Fused 18F-FDG PET/CT and MIP images showed increased metabolism in aortic wall and supraortic vessels which corresponded to vasculitis. Increased 18F-FDG uptake in peripheral joints was also described with shoulder involvement. Inflammatory activity in the wrists was also observed, as a characteristic arthritis in this pathology.

bursa (Fig 2). Of the 23 patients with joint uptake in the $^{18}$FDG-PET/CT scan, the area most frequently affected was the sternoclavicular (23.28%), followed by the knees (10.95%) and the wrist (8.21%).

Of the 16 patients diagnosed with LVV, in addition to glucidic uptake increase in the aortic wall, $^{18}$FDG PET/CT scans showed a concomitant uptake in the shoulders, hips and bursae. We did not measure vascular diameters in the patients with PMR included in the study. Nevertheless, in the visual evaluation, we did not observe dilated vessels. Table 3 shows the sites of $^{18}$FDG accumulations in patients with PMR and LVV.

## Malignancies

Of the five patients who were finally diagnosed with neoplastic processes (four haematological and one transverse colon) and who first presented with PMR, the $^{18}$FDG-PET/CT scan showed

**Table 2. TBR of the different locations according to the final diagnosis.**

| Final diagnosis | TBR _shoulder | TBR _column | TBR _joint | TBR-aorta | TBR_bursae |
|---|---|---|---|---|---|
| PMR | 2.03±0.9 | 1.78±0.9 | 2.74±1.09 | 1.52±0.217 | 2.42±0.91 |
| LVV | 1.80±0.55 | 1.95±1.05 | 2.51±0.77 | 1.98±0.64 | 2.77±1.07 |
| NEOPLASIA | 1.60±0.69 | 1.40±0.67 | 1.58±. | 1.46±0.22 | 1.33±. |
| OTHER | 2.38±0.94 | 1.90±0.79 | 3.12±2.07 | 1.92±0.42 | 3.085±0.6 |
| p | P = 0.370 | P = 0.472 | P = 0.516 | P = <0.001 | P = 0.104 |

TBR_bursae: Includes trochanteric and subacromial bursitis.

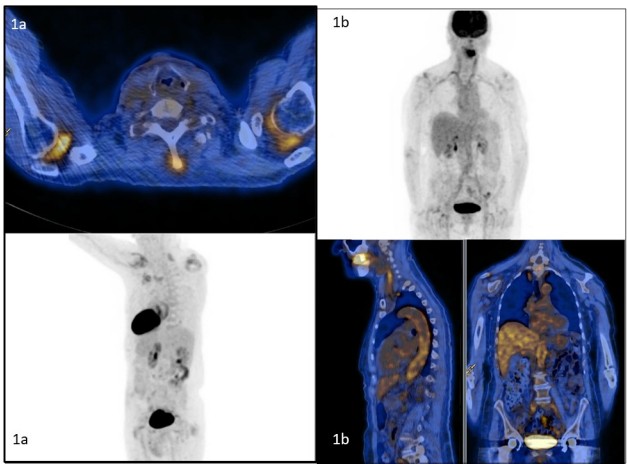

**Fig 2. <sup>18</sup>FDG-PET/CT images.** 1A: Uptake in shoulders and cervical interspinous bursae.1B: Uptake in supra- and infra-diaphragmatic aorta.

accumulations suggestive of neoplastic process in three patients (mediastinal lymphadenopathy and increased diffuse glucidic metabolism in bone marrow in two patients with lymphoproliferative syndrome (Fig 3), along with a hypermetabolic lesion in the transverse colon with the diagnosis of a high-grade villous adenoma in the third patient). The $^{18}$FDG-PET/CT scan thus helped diagnose three of the 103 patients who had neoplastic processes.

**Table 3. Distribution of glycemic metabolism intake in patients diagnosed with polymyalgia rheumatica (PMR) and large vessel vasculitis (LVV).**

| Glycemic metabolism intake in PMR and/or Vasculitis | N (%) |
| --- | --- |
| **PMR** | 73 (70.9) |
| Shoulder involvement | 40 (54.8) |
| Hip involvement | 27 (37.0) |
| Peripheral joints | 23 (31.5) |
| • Sternoclavicular | 17 (23.2) |
| • Knee | 6 (8.2) |
| • Wrist | 8 (10.9) |
| Bursae uptake | 34 (46.6) |
| • Bursae cervical spine | 22 (30.1) |
| • Bursae dorsal spine | 3 (4.1) |
| • Bursae lumbar spine | 25 (34.2) |
| • Ischiatic bursae | 26 (35.6) |
| **LVV** | **16 (15.5)** |
| Shoulder involvement | 9 (56.3) |
| Hip involvement | 6 (37.5) |
| Bursae uptake | 11 (68.8) |
| • Bursae cervical spine | 9 (56.3) |
| • Bursae lumbar spine | 11 (68.8) |
| • Ischiatic bursae | 6 (37.5) |

Table shows distribution of glycemic metabolism intake in all patients in the sample diagnosed with PMR (n = 73) or LVV (n = 16).

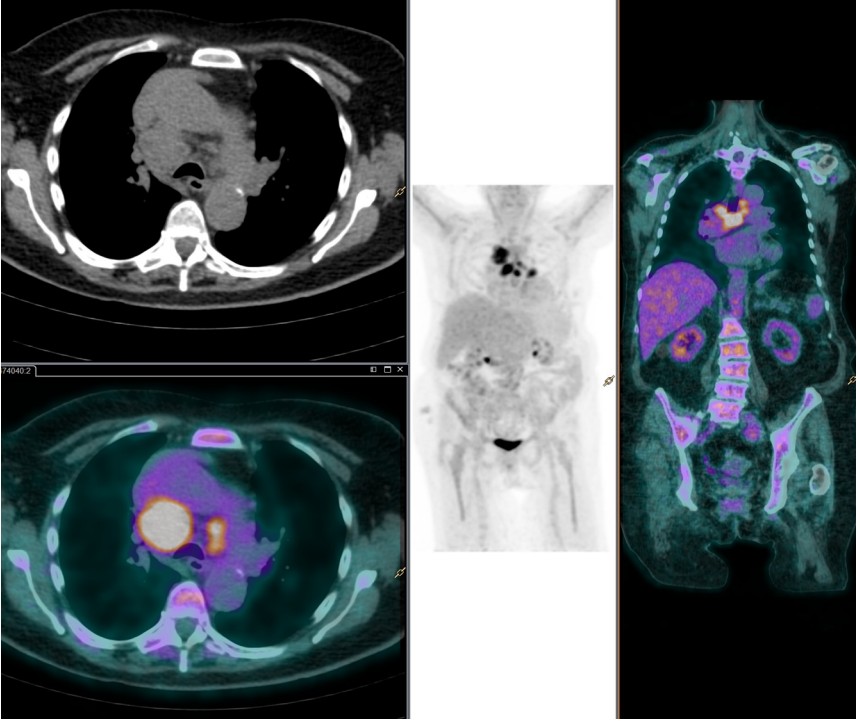

**Fig 3. [18]FDG-PET/CT images.** A 89-year-old woman 18F-FDG PET/CT with diagnostic of polimyalgia rheumatica. PET/CT images showed multiple mediastinal nodules with increase [18]F-FDG uptake and bone marrow glucose hypermetabolism. The patient was finally diagnosed of in a lymphoproliferative process. No treatment was administered due to age and other concomitant pathologies.

## [18]FDG-PET/CT in onset PMR versus PMR refractory to glucocorticoids

When we performed the statistical analysis and compared the clinical variables of patients with PMR_os and patients with PMR_sr, we observed that the former presented more asthenia, weight loss and fever than the latter group. We did not find differences in waist pain, cervicalgia or analytical parameters. Table 4 shows the statistical differences observed in patients with PMR_os and patients with PMR_sr.

We classified the final diagnoses into four groups: PMR, LVV, neoplasms and other diagnoses (including PMR with EORA, Sjögren syndrome, small vessel vasculitis and a degenerative process). When we compared these final diagnoses considering whether they were PMR_os or PMR_sr, we found significant differences in the final diagnosis (p = 0.045) (Table 5).

## Discussion

The present study was designed to rule out a wide spectrum of differential diagnoses in a group of patients studied by [18]FDG-PET/CT. We included 103 patients with an initial diagnosis of PMR. The confirmed diagnosis in our cohort was PMR in 70.9%, LVV in 15.5%, neoplasms in 4.8%, and other diagnoses in the remaining 8.8%.

Several studies corroborate the already known relationship between PMR and LVV studied by PET/CT scan. But unlike our study, the main objective of most of these studies was not to analyse the wide range of differential diagnoses presented by the PMR but to point out certain specifities of LVV and PMR. To date, few articles have addressed the issue of the broad range of differential diagnoses presented by PMR by PET/CT scan.

**Table 4. Clinical features of patients: a) with onset PMR vs steroid resistant PMR b) with the final diagnosis of PMR, LVV or malignancy.**

|  | PMR global | LVV | Malignancies | PMR_os | PMR_sr | p |
|---|---|---|---|---|---|---|
| Women, n (%) | 23 (31.5%) | 4 (25%) | 2 (40.0%) | 33 (63.5%) | 40 (78.4%) | 0.129 |
| Asthenia, n (%) | 49 (67.1%) | 11 (68.8%) | 3 (60.0%) | 40(76.9%) | 26 (51.0%) | **0.006**\* |
| Weight loss, n (%) | 21 (28.7%) | 10 (62.5%) | 4 (80.0%) | 25 (48.1%) | 13 (25.5%) | **0.017**\* |
| Fever, n (%) | 11 (15.1%) | 2 (12.5%) | 0 (0%) | 12 (23.1%) | 4 (7.8%) | **0.029**\* |
| Morning stiffness, n (%) | 38 (52.1%) | 6 (37.5%) | 2 (40.0%) | 27 (51.9%) | 22 (43.1%) | 0.37 |
| Cervicalgia, n (%) | 52 (71.2%) | 12 (75%) | 2 (40.0%) | 34 (65.4%) | 37 (72.5%) | 0.432 |
| Shoulder pain, n (%) | 71 (97.3%) | 15 (93.8%) | 5 (100%) | 51 (98.1%) | 49 (96.1%) | 0.543 |
| Hip pain, n (%) | 66 (90.4%) | 11(68.8%) | 5 (100%) | 48 (92.3%) | 41 (80.4%) | 0.074 |
| Headache, n (%) | 13 (17.8%) | 7 (43.8%) | 1 (20.0.%) | 11 (21.2%) | 12 (23.5%) | 0.772 |
| Amaurosis, n (%) | 1 (1.4%) | 4 (25%) | 0 (0%) | 2 (3.8%) | 3 (5.9%) | 0.630 |
| Jaw claudication, n (%) | 4 (5.5%) | 5 (31.3%) | 0 (0.0%) | 4 (7.7%) | 5 (9.8%) | 0.704 |
| Reactive C protein (mg/L), mean ± SD | 29.7±46.1 | 43.3±45.3 | 48.9±72.17 | 44.8 ± 59.5 | 31.1 ± 78.3 | 0.287 |
| Erythrosedimentation (mm/s), mean ± SD | 53.2±3.1 | 69.3±28 | 59±39.8 | 59.3 ± 29.4 | 52.4 ± 32.5 | 0.663 |
| Hemoglobin (g/L), mean ± SD | 120.3±25.4 | 116.8±12.1 | 118±9.01 | 117.4 ± 23.2 | 120.0 ± 26.7 | 0.880 |
| Prednisone dose, mean ± SD | 11.5±11.64 | 11.0±12.2 | 6.6±11.5 | 10.7 ± 11.9 | 11.4 ± 10.8 | 0.124 |

PMR_os: Onset PMR patients; PMR_sr: Steroid resistant PMR patients; PMR global: PMR_os and PM_sr, LVV: Large vessel vasculitis.

Henckaerts et al. [13] prospectively included 99 consecutive patients with a possible clinical diagnosis of PMR. All patients underwent $^{18}$FDG-PET scanning before treatment with steroids. As in our study, the gold standard for a diagnosis of PMR was the judgment of an experienced clinician. A final diagnosis of isolated PMR was made in 67.6% of the patients (as in our study), while another condition was diagnosed in the remaining 32.32% of patients. Diagnoses made in non-PMR patients were for a variety of diseases that canpresent with similar clinical symptoms.

## Malignancies

In our cohort, five patients (4.8%) presented neoplastic processes (four hematological neoplasms and one transverse colon high-grade villous adenoma). The $^{18}$FDG-PET/CT scan helped diagnose malignant processes in three of the 103 patients included. Published data to date on this topic are scarce. In one recent paper, in a cohort of 99 patients with PMR, Henckaerts L et al. [13] found one patient who presented a paraneoplastic syndrome secondary to a carcinoid neoplasm (visible on PET scan). In another study, Palard Novello et al. [14] studied 21 patients with "new onset PMR" in order to evaluate the use of the $^{18}$FDG-PET/CT scan in assessing tocilizumab treatment and found that one patient presented a malignant process in the $^{18}$FDG-PET/CT scan pre-treatment. In a large database study from the Swedish Hospital Discharge Register, Ji J et al. [16] included 5,918 patients with GCA and PMR and reported a clear increased risk of cancer within the first year of PMR diagnosis. Similarly, Muller et al.

**Table 5. Final diagnosis of patients with onset PMR (PMR_os) vs steroid resistant (PMR_sr).**

| Final diagnosis | PMR | LVV | NEOPLASIA | OTHER |
|---|---|---|---|---|
| PMR_os, n (%) | 37 (50.7) | 6 (37.5) | 5 (100) | 4 (44.4) |
| PMR_sr, n (%) | 36 (49.3) | 10 (62.5) | 0 (0) | 5 (55.6) |
| Total | 73 | 16 | 5 | 9 |

p = 0.045. PMR: Polymyalgia rheumatica; LVV: Large vessel vasculitis.

[15] described a 69% increased risk of malignancy in patients with PMR within the first six months of diagnosis. One explanation for the high risk observed in their study may be that the high rate of health care consumption allowed for a higher rate of cancer detection. In contrast with the results of Ji J et al, several studies found no association [14–19] with an increase in the risk of cancer after PMR diagnosis.

It is difficult to reach a firm conclusion or confirm this association because most studies included have a referral bias, a lack of control groups, and/or a limited follow up period. Although the association between PMR and malignant processes is well known, results from published articles and reviews do not offer consistent or strong evidence. Our data show a non-significant rate of malignant processes, but we cannot definitely conclude whether or not this rate is higher than that in the population without PMR because we did not have a control group.

## Large vessel vasculitis

Multiple published papers describe the relationship between LVV and PMR studied by PET-CT, but the range of percentages of this association varies widely. For instance, Henckaerts et al. [13] described associations of around 2%, which is very different from the 65% described by Lavado-Perez et al. and Prieto-Peña et al. [20,21]. Prieto-Peña studied 84 patients with classic PMR. A PET/CT scan was positive for LVV in 51 (60.7%) patients. The differences observed in the incidence of LVV described by Prieto-Peña [21] compared to our results could be explained by the differences in the populations studied. Prieto et al's patients were all steroid resistant. When classifying the patients in our sample as steroid-resistant or onset we observed that the frequency of LVV in the PMR_sr was 62.5% compared to 37.5% in the PMR_os group, suggesting a consistency with the results of the Prieto-Peña study.

Prieto-Peña also studied whether there was an association between LVV and clinical and analytical variables. They described that pelvic girdle pain, inflammatory low back pain and lower limb pain were predictors of a positive [18]FDG-PET/CT scan result for LVV in patients with PMR. However, they did not find an association with any analytical parameters or with the presence of constitutional symptoms. Like Prieto- Peña, we found no-significant association between LVV and PMR in patients who presented analytical alterations. Conversely, Gonzalez Gay et al. [22] reported that patients with PMR associated with GCA had significant alterations in erythrocyte sedimentation rate, platelet count and hemoglobin compared to patients with isolated PMR.

Furthermore, as in Prieto-Peña et al's study., the work of Lavado-Perez et al. [20] described 40 patients diagnosed with PMR using PET/CT scan who showed a high prevalence of LVV (65%), which is considerably higher than our prevalence. Once again, this difference could be explained by the fact that 33 of their 40 patients had suspected LVV before [18]FDG-PET/CT, strengthening the notion that despite being PMR, the cohorts of patients studied and published to date with [18]FDG-PET/CT scan are not homogenous.

In contrast, Henckaerts et al. [13] reported fewer cases of LVV than those described by previous studies. They found that only 15% of PMR patients had an increased FDG uptake in the larger thoracic vessels, compared with 6% of control patients. Of the patients with a vascular [18]FDG uptake, two were diagnosed with GCA based on a positive temporal artery biopsy. They attributed these findings to blood pool activity or no activity in the vessel wall, and to defects in the technique in the collection of images in some of the scans.

Concerning TBR aorta data, in our cohort, the prevalence of TBR of the aorta in the patients included in the "other group" was similar to that in patients in the LLV group. There may be several plausible explanations for this. The first is the small size and heterogeneity of

the "other group". Furthermore, the final diagnosis was based on [18]FDG-PET/CT results and the presence of additional symptoms (clinical, analytical and evolutionary data). For these reasons the patients were not labelled LVV.

Taken together, it is evident that the frequency of association between PMR and LVV differs significantly depending on the characteristics of the cohorts studied. The time of disease evolution, the treatment administered and the presence of new symptoms of LVV are variables that must be considered when describing the association between PMR and LVV studied by [18]FDG-PET/CT.

## Distribution of increased glucidic metabolism in [18]FDG-PET/CT scan

In our study, the majority of patients diagnosed with PMR showed an increase in glucidic metabolism in the shoulders, followed by hips and peripheral joints. Almost half of the patients showed uptake in a bursa. The most frequently affected peripheral areas were the sternoclavicular, followed by the carpus and the knees.

In a restrospective study of 50 PMR patients undergoing a [18]FDG-PET/CT scan, Sondag et al. [23] found greater uptake in the shoulders than in the hips, with percentages of distribution that were very similar to ours (54% and 36% for shoulders and hips, respectively). Two other authors, Rehak et al. [24] and Henckaerts et al. [13], also observed greater uptake in the shoulders than in the hips, though their percentages were considerably higher than ours. Rehak et al. evaluated 67 patients and found that 87% and 70%, respectively, of the patients showed uptake in the shoulders and hips. Henckaerts et al. evaluated 67 patients and also described a greater uptake in the shoulders than the hips than in our study (97% in shoulders and 67% in hips compared with our figures of 54.8% and 37%, respectively). This higher percentage of hip and shoulder involvement in Rehak's and Henckaert's studies [13,24] compared to our findings, as well as those reported by Sondag [23], could be because no patients in their study were undergoing steroid treatment.

We also analyzed the [18]FDG-PET/CT scan uptake frequency in bursae. We observed a 35.6% uptake in the ischial bursa in patients with PMR, frequencies that are lower than those described by the groups in the studies by Rehak [24], Sondag [22] and Henckaerts [13], who found an uptake from 52% to 67% in patients with PMR. Likewise, we observed uptake in 30.1% and 34.2% of cervical and lumbar spinous processes, respectively. We also found a similar frequency and involvement when comparing lumbar and cervical processes. Rehak and Sondag found a greater involvement in lumbar processes (57% and 38% respectively) than in cervical processes (19.5% and 10% respectively). Interestingly, we found that dorsal processes were involved in 4.1% of patients, a finding that had not been described previously.

Our results showed peripheral joint uptake. This is controversial considering the possible differential diagnosis with EORA in patients with PMR. Although we describe 8.2% of involvement in knees and 10.95% in wrists, the final diagnosis of PMR stands, though not that of EORA. A few authors describe peripheral joint uptake in detail in patients with PMR [25,26]. Kaneko et al. [27] studied 20 patients with PMR, describing an exceptionally high frequency of knee involvement (96.2%). However, only four patients complained of knee symptoms. The same group also described a high frequency of involvement of the wrists, with figures of up to 40%. Cimmino et al. [28] reported a higher frequency of knee involvement (84%) in a study using [18]FDG-PET/CT. On the other hand, in 2018, Yuge et al. [25] studied 16 patients with a definitive diagnosis of PMR and described a significant incidence (6%) of [18]FDG uptake in the wrists. Considering the small number of patients in these previous studies, it is difficult to reach a firm conclusion concerning peripheral uptake. We still need to define those findings in pure PMR and observe whether they can develop into EORA over time.

In our cohort, FDG-PET was helpful to rule out vasculitis, malignancy and peripheral arthritis. Nevertheless, the TBRs in the typical PMR locations (spine/shoulders/hips) did not differ significantly from those in the"other diagnosis" group. TBR at typical PMR locations can be similar in groups for two reasons. First, other diagnoses, such as LVV and neoplasms, can be accompanied by PMR, and second, the group of "other diagnoses" is small, making it more difficult to find statistically significant differences.

## Limitations

This study has several limitations. The main limitation is its retrospective character. Nevertheless, our results clearly reflect observations in our daily clinical practice. Second, as some patients were under treatment with steroids at the time of the [18]FDG-PET/CT scan, our results cannot be compared statistically to studies involving steroid-naïve patients only. A third limitation is the lack of a control group. However, we would not have been able to compare the diagnosis found in patients with PMR using a [18]FDG-PET/CT scan with the general population for ethical and procedural reasons. Fourth, it has been suggested that a late PET scan at 180 minutes increase the accuracy of the method in diagnosing LVV, especially for the evaluation of the thoracic aorta; but unfortunately this scan is not available in our cohort. Fifht, the inclusion of patients who concomitantly presented GCA symptoms may entail a bias.

## Conclusion

Based on our findings, we consider the [18]FDG-PET/CT scan to be a very useful tool, not only to help diagnose PMR, but also to identify diseases such as LVV and malignant processes that are associated with this disorder. However, the best moment to perform this technique (PMR_os or PMR_sr) remains to be defined.

## Supporting information

**S1 File.**
(PDF)

## Author Contributions

**Conceptualization:** Patricia Moya-Alvarado, Alejandro Fernandez Leon, Maria Emilia Corica.

**Data curation:** Patricia Moya-Alvarado, Alejandro Fernandez Leon, Maria Emilia Corica, Valle Camacho Marti, Diego Alfonso López-Mora.

**Formal analysis:** Patricia Moya-Alvarado, Diego Alfonso López-Mora, Ivan Castellví.

**Investigation:** Patricia Moya-Alvarado, Alejandro Fernandez Leon.

**Methodology:** Patricia Moya-Alvarado, Alejandro Fernandez Leon.

**Supervision:** Ivan Castellví, Hèctor Corominas.

**Validation:** Alejandro Fernandez Leon, Hèctor Corominas.

**Writing – original draft:** Patricia Moya-Alvarado, Hèctor Corominas.

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
