## [Decision Letter · Decision Letter 0]

19 May 2021

PONE-D-21-12919

THE ADDED VALUE OF 18F-FDG PET/CT IN THE ASSESSMENT OF ONSET AND STEROID RESISTANT POLIMYALGIA RHEUMATICA

PLOS ONE

Dear Dr. Moya,

Thank you for submitting your manuscript to PLOS ONE. After careful consideration, we feel that it has merit but does not fully meet PLOS ONE’s publication criteria as it currently stands. Therefore, we invite you to submit a revised version of the manuscript that addresses the points raised during the review process.

We look forward to receiving your revised manuscript.

Kind regards,

Pierpaolo Alongi

Academic Editor

PLOS ONE

Additional Editor Comments:

The article regards an interesting topic. The design of the study is good as the results presented. The paper could be accepted after minor revision.

Journal Requirements:

3. Thank you for including your ethics statement:

“The study was approved by the local institutional ethics committee (IIBSP-VAS-2013-122) and all

patients signed an informed consent form before enrollment."

Reviewers' comments:

Reviewer's Responses to Questions

**Comments to the Author**

1. Is the manuscript technically sound, and do the data support the conclusions?

Reviewer #1: Yes

Reviewer #2: Yes

2. Has the statistical analysis been performed appropriately and rigorously? 

Reviewer #1: Yes

Reviewer #2: I Don't Know

3. Have the authors made all data underlying the findings in their manuscript fully available?

Reviewer #1: Yes

Reviewer #2: Yes

4. Is the manuscript presented in an intelligible fashion and written in standard English?

Reviewer #1: Yes

Reviewer #2: Yes

5. Review Comments to the Author

Reviewer #1: The authors described the findings of 18FDG-PET/CT in a cohort of patients with PMR and detailed how the PET findings increased the accuracy in the diagnosis of other underlying conditions.

The work is well structured and written fluently; the methods are well illustrated and clear and the data are adequately described.

The results confirm the close association between PMR and LVV and strengthen the indication for the use of FDG-PET in this group of patients to obtain a more accurate characterization.

Comments:

1. a late PET scan at 180 minutes was suggested to increase the accuracy of the method in diagnosing LVV, especially for the evaluation of the thoracic aorta; as the study is retrospective it is understandable that this scan is not available, but this should be indicated in the limitations.

2. three out of five patients with malignancy showed positive PET. It would be interesting to know the time elapsed between PET scan and tumor diagnosis in the two patients who tested negative; it could be tumors that arose some time later and which, consequently, were not diagnosable at the time of the examination

3. some authors have also suggested a qualitative evaluation of the degree of vascular uptake of FDG in case of vasculitis compared to the degree of uptake of the liver; it would also be interesting to know the results of this type of analysis compared to what is described by the semi-quantitative method

4. there are rare typos to correct: "introduction", sixth line "andit" instead of "and it"; "limitations", fifth line "ableto" instead of "able to"

Reviewer #2: Congratulations to the authors for the choice of the topic, undoubtedly interesting, and for the work done. The purpose of the work is clearly defined. Materials and methods are correctly described as well as the results reported in an adequate number of tables. Moreover, the limitations of the work are well reported before the conclusions, and one of the most important is obviously the lack of a control Group.

Nevertheless, some points deserve more attention:

1. a better definition of the positivity criteria for vasculitis (Meller visual score?, a semiquantitative analysis with a defined cutoff value?, other?) and of vascular segments involved (for example, if there is a prevalence of a specific aortic district in patients affected by PMR).

2. to include "FDG-PET/CT (A) imaging in large vessel vasculitis and polymyalgia rheumatica: joint procedural recommendation of the EANM, SNMMI, and the PET Interest Group (PIG), and endorsed by the ASNC" among the cited references.

3. at least two additional figures related to cited "neoplasm" and "other diagnosis"are needed.

6. PLOS authors have the option to publish the peer review history of their article (what does this mean?). If published, this will include your full peer review and any attached files.

Reviewer #1: No

Reviewer #2: No

---

## [Author Response · Author response to Decision Letter 0]

2 Jul 2021

Comments to the Author

Review Comments to the Author

Reviewer #1: The authors described the findings of 18FDG-PET/CT in a cohort of patients with PMR and detailed how the PET findings increased the accuracy in the diagnosis of other underlying conditions.

The work is well structured and written fluently; the methods are well illustrated and clear and the data are adequately described.

The results confirm the close association between PMR and LVV and strengthen the indication for the use of FDG-PET in this group of patients to obtain a more accurate characterization. Thanks for your comment

Comments:

1. A late PET scan at 180 minutes was suggested to increase the accuracy of the method in diagnosing LVV, especially for the evaluation of the thoracic aorta; as the study is retrospective it is understandable that this scan is not available, but this should be indicated in the limitations.

As the reviewer describes, it has been suggested in the literature that a late PET scan at 180 minutes increases the accuracy of the method in diagnosing LVV, especially for the evaluation of the thoracic aorta. Unfortunately, this scan is not available in our cohort while it wasn´t planned from the beginning. The usefulness of the PET Scan has presented nice improvements in the recent years not only in the technique accuracy, but also on its indications and time of performance. We have added this suggestion in the limitations section.

2. Three out of five patients with malignancy showed positive PET. It would be interesting to know the time elapsed between PET scan and tumor diagnosis in the two patients who tested negative; it could be tumors that arose some time later and which, consequently, were not diagnosable at the time of the examination

The other two patients who were diagnosed with neoplastic processes in whom the 18FDG-PET/CT showed no accumulations suggestive of neoplastic process, were diagnosed finally with Chronic Lymphatic Leukemia in stage A0 and a Marginal zone B-cell lymphoma (MZL) with peripheral expression. In both cases, the 18FDG-PET/CT scan showed no evidence of a neoplastic process, and they were diagnosed of hematological disorders (lymphocytosis, and anemia with leukocytosis and disrupted immunophenotype, respectively). Both neoplastic processes were diagnosed in parallel temporally with the 18FDG-PET/CT scan. 

We have included these clarifications in the corresponding section.

3. Some authors have also suggested a qualitative evaluation of the degree of vascular uptake of FDG in case of vasculitis compared to the degree of uptake of the liver; it would also be interesting to know the results of this type of analysis compared to what is described by the semi-quantitative method

In this study we performed both hepatic and vascular (cava vein) measurements in order to calculate TBR values (target–to–background ratio). Results were quite similar and we aimed to divide the vascular wall SUV by the venous blood pool SUV to correct for blood uptake. As it is proposed in "FDG-PET/CT (A) imaging in large vessel vasculitis and polymyalgia rheumatica: joint procedural recommendation of the EANM, SNMMI, and the PET Interest Group (PIG), and endorsed by the ASNC", the normalization of the arterial wall uptake to the background activity of venous blood pool, also grading of arterial inflammation against the liver background are both established method that provide a good reference for assessing vascular inflammation. 

4. there are rare typos to correct: "introduction", sixth line "andit" instead of "and it"; "limitations", fifth line "ableto" instead of "able to"

We apologize for that and we have made the changes of the rare typos indicated and carefully read and reviewed the entire document again.

Reviewer #2: Congratulations to the authors for the choice of the topic, undoubtedly interesting, and for the work done. The purpose of the work is clearly defined. Materials and methods are correctly described as well as the results reported in an adequate number of tables. Moreover, the limitations of the work are well reported before the conclusions, and one of the most important is obviously the lack of a control Group. T 

We thank you very much for these nice comments and considerations

Nevertheless, some points deserve more attention:

1. a better definition of the positivity criteria for vasculitis (Meller visual score?, a semiquantitative analysis with a defined cutoff value?, other?) and of vascular segments involved (for example, if there is a prevalence of a specific aortic district in patients affected by PMR).

We performed 18FDG PET/CT visual analysis by two experimented Nuclear medicine physicians in order to determine positivity in vasculitis. We did not use visual scores in this retrospective study because each examination was reported as positive or negative before the analysis. However, we previously performed a comparative work studying 59 patients diagnosed with PMR and suspected vasculitis and 30 oncologic patients (as controls). 

We calculated Target to Background Ratios and we compared TBR values among PMR patients with suspected vasculitis, without, and controls. We also calculated ROC curves for specificity and sensibility in order to find a TBR cutoff value useful for potential diagnosis. The results showed that the TBR cutoff value that provided better discrimination amount groups was 1.26 with the best sensibility (0.83) and specificity (0.93) values. How this is out of the scope of this paper, we may better understand the initial findings and therefore, we will submit for consideration our results included in another project.

We have included these clarifications in the corresponding section.

2. to include "FDG-PET/CT (A) imaging in large vessel vasculitis and polymyalgia rheumatica: joint procedural recommendation of the EANM, SNMMI, and the PET Interest Group (PIG), and endorsed by the ASNC" among the cited references.

We have added the proposed reference in bibliographic section.

3. at least two additional figures related to cited "neoplasm" and "other diagnosis"are needed.

As the reviewer suggests, we have added two more images, one of lymphoproliferative process and the other of arthritis diagnoses.

---

## [Editor Report · Decision Letter 1]

12 Jul 2021

THE ADDED VALUE OF 18F-FDG PET/CT IN THE ASSESSMENT OF ONSET AND STEROID RESISTANT POLIMYALGIA RHEUMATICA

PONE-D-21-12919R1

Dear Dr. Moya,

We’re pleased to inform you that your manuscript has been judged scientifically suitable for publication and will be formally accepted for publication once it meets all outstanding technical requirements.

Kind regards,

Pierpaolo Alongi

Academic Editor

PLOS ONE
---

## [Editor Report · Acceptance letter]

15 Sep 2021

PONE-D-21-12919R1 

The added value of 18f-FDG PET/CT in the assessment of onset and steroid resistant polimyalgia rheumatica 

Dear Dr. Moya:

I'm pleased to inform you that your manuscript has been deemed suitable for publication in PLOS ONE. Congratulations! Your manuscript is now with our production department. 

Kind regards, 

on behalf of

Dr. Pierpaolo Alongi 

Academic Editor

PLOS ONE